# Predicting treatment benefit in multiple myeloma through simulation of alternative treatment effects

Joske Ubels [1,2,3], Pieter Sonneveld[2], Erik H. van Beers[3], Annemiek Broijl[2], Martin H. van Vliet[3] & Jeroen de Ridder [1]

Many cancer treatments are associated with serious side effects, while they often only benefit a subset of the patients. Therefore, there is an urgent clinical need for tools that can aid in selecting the right treatment at diagnosis. Here we introduce simulated treatment learning (STL), which enables prediction of a patient's treatment benefit. STL uses the idea that patients who received different treatments, but have similar genetic tumor profiles, can be used to model their response to the alternative treatment. We apply STL to two multiple myeloma gene expression datasets, containing different treatments (bortezomib and lenalidomide). We find that STL can predict treatment benefit for both; a twofold progression free survival (PFS) benefit is observed for bortezomib for 19.8% and a threefold PFS benefit for lenalidomide for 31.1% of the patients. This demonstrates that STL can derive clinically actionable gene expression signatures that enable a more personalized approach to treatment.

[1] Department of Genetics, Center for Molecular Medicine, University Medical Center Utrecht, Universiteitsweg 100, 3584 CG Utrecht, The Netherlands. [2] Department of Hematology, Erasmus MC Cancer Institute, Wytemaweg 80, 3015 CN Rotterdam, The Netherlands. [3] SkylineDx, Marconistraat 16, 3029 AK Rotterdam, The Netherlands. Correspondence and requests for materials should be addressed to M.Vliet. (email: m.vanvliet@skylinedx.com) or to J.Ridder. (email: j.deridder-4@umcutrecht.nl)

The successful treatment of cancer is hampered by genetic heterogeneity of the disease. Differences in the genetic makeup between tumors can result in a different response to treatment[1]. As a result, despite the existence of a wide range of efficient cancer treatments, many therapies only benefit a minority of the patients that receive them[2]. Because many therapies may be associated with serious adverse effects, there is a great clinical need for tools to predict—at the moment of diagnosis—which patient will benefit most from a certain treatment.

To address this, substantial efforts have been made to identify clinical and molecular markers, such as gene expression signatures, that can predict a favorable or adverse prognosis[3]. Traditionally, this is achieved by defining subtypes (e.g., through unsupervised learning approaches) based on molecular markers such as genotype or gene expression. For many of these subtypes an association has been determined to survival or drug response[4–6].

More direct approaches use supervised learning, such as (logistic) regression, to identify markers associated with survival. In this setting, a class label is defined for each patient based on their survival or some other outcome measure, such as the risk of experiencing a relapse. The training procedure then focuses on predicting these labels as accurately as possible to ultimately produce a classifier that can predict outcome for a new patient. One of the first successful examples of such an approach resulted in a 70-gene prognostic expression signature for breast cancer[7]. A phase III clinical trial recently revealed that patients predicted to have good survival based on this signature can safely forego chemotherapy without compromising outcome[8], thus preventing overtreatment of these patients. These examples demonstrate that prognostic predictors can have value in predicting benefit to treatment.

Despite these successes, prognostic signatures are fundamentally limited in their ability to predict treatment benefit. This is because prognostic signatures are determined without taking treatment into account, i.e., they are not trained to distinguish patients that survive long as a result of the treatment. For this reason, patients classified in the "long survival" class may in fact survive just as long on any treatment available. Conversely, patients in the "short survival" class could actually have benefited from the treatment because they would have had an even shorter survival on another treatment. In Fig. 1a, b, we illustrate this in the setting of a randomized trial with two treatment arms. Figure 1a shows the result for a prognostic classifier which results in a survival difference between the two classes that is similar in both treatment arms. However, to achieve treatment benefit prediction we should identify a subset of patients that specifically benefit from one of the two treatments, that is, where the difference in survival between the two treatments is larger than in the population as a whole (Fig. 1b). It should be noted that it is possible that a prognostic classifier happens to identify a difference between treatment arms as well, but this is not an aim in the training procedure. We hypothesized that a method that is specifically geared towards optimizing the identification of a subset of patients with a greater treatment benefit will achieve better results.

Treatment benefit is commonly measured by the hazard ratio (HR), which describes a patient's hazard to experience an event, for example death or progression of disease, relative to another set of patients who received a different treatment. Some recently published predictive classifiers have only shown to find a difference in response or survival between two groups of patients who all received the same treatment[9–11]. These signatures are not constructed to be predictive, since they do not necessarily provide a treatment decision; the prognosis may well be the same in every treatment group. To be truly predictive, a subgroup with a difference in survival between two treatment arms needs to be identified.

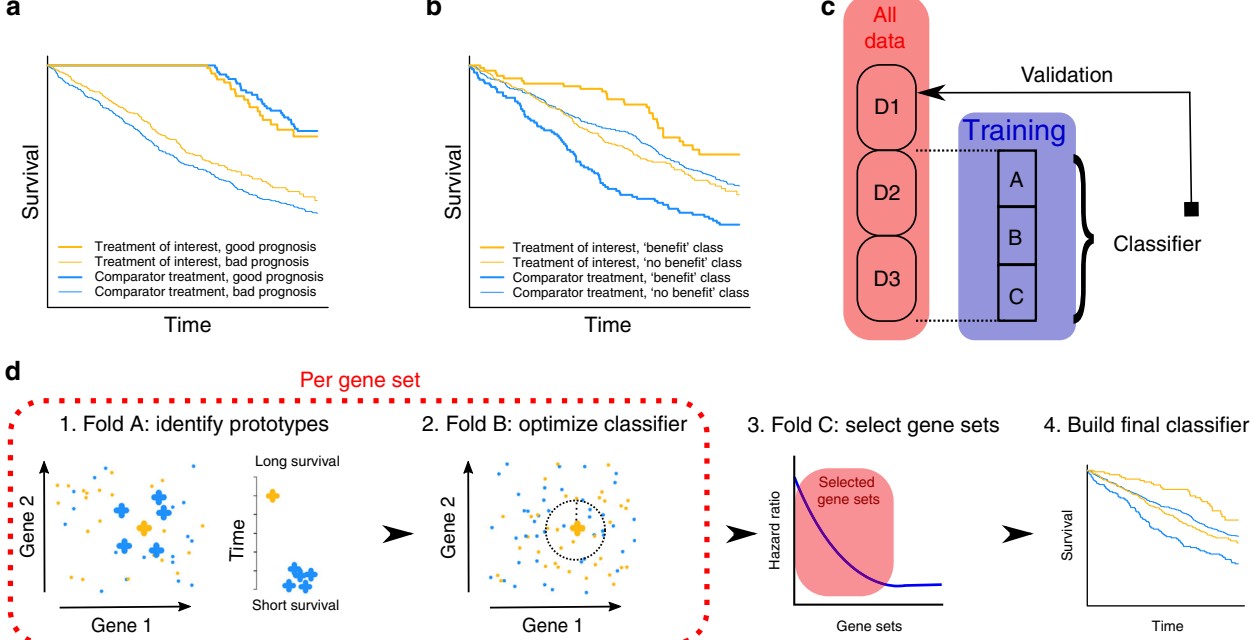

**Fig. 1** Illustration of the difference between prognostic and predictive classifiers and an overview of the approach. **a** Example of the Kaplan–Meier curve for a prognostic classifier. **b** Example of the Kaplan–Meier curve for a predictive classifier. **c** Division of dataset into training and test sets. D1–D3 are all used once to validate the classifier trained on the remaining two-thirds of data. **d** Flow of the GESTURE algorithm. In step 1 the prototypes with a longer than expected survival difference are identified on fold A. In step 2 the number of prototypes and corresponding decision boundary used in the classifier are optimized on fold B. In step 3 the performance of the classifier on fold C across all repeats is used to select the combination of gene sets to be used in the final classifier. In step 4 a classifier for these gene sets is defined on all training data. This classifier will be validated on the fold D not included in the training data

Constructing classifiers that can achieve true treatment benefit prediction thus poses a unique challenge, as it is impossible to know how a patient would have responded to the alternative treatment. As a result, class labels based, which can be used to train a classifier are not available and existing classification schemes are not applicable (as demonstrated in Sections Results and Discussion).

To address the lack of suitable training labels, we introduce the concept of simulated treatment learning (STL), a method to derive classifiers that can predict treatment benefit. STL can be applied to gene expression datasets with two treatment arms and survival data. STL uses genetic similarity, defined based on gene expression in the tumor, between patients from different treatment groups to model how a particular patient would have responded to the alternative treatment.

In this work we focus on predicting treatment benefit for multiple myeloma (MM), a clonal B-cell malignancy that is characterized by abnormal proliferation of plasma cells in the bone marrow. Median survival of MM patients is 5 years[12]. In the last two decades many novel therapies have been introduced for MM, resulting in an improved survival[13,14]. Bortezomib and lenalidomide were crucial in achieving these improved survival rates. However, despite these advances, not all patients benefit from these novel agents and there are insufficient tools to predict treatment response or survival. Between MM patients heterogeneity in gene expression profiles is observed[15,16]. For these reasons, genetic signatures that can predict treatment benefit for MM patients are of high clinical value, making it an ideal test case for STL.

There are some preliminary indications that predictive signatures may exist for MM. Some of the various prognostic factors known in MM were later found to be predictive as well. For instance, it was shown that patients with the chromosomal aberration del17p, known to be prognostic, benefitted more from the proteasome inhibitor bortezomib than patients without del17p[17]. Furthermore, expression levels of tumor suppressor *RPL5*, located on chromosome 1, were also found to correlate with bortezomib response[18]. Both these abnormalities have been found to be recurrently present in MM plasma cells and were later found to be prognostic and predictive. STL enables us to directly discover predictive markers, without relying on previously discovered (prognostic) markers. We implement the STL concept in the algorithm Gene Expression-based Simulated Treatment Using similaRity between patiEnts (GESTURE), which makes it possible to derive a gene expression signature that is able to distinguish a subset of patients with improved treatment outcome from the treatment of interest, but not from the comparator treatment. We show that GESTURE can predict treatment benefit for two major treatments in MM, bortezomib and lenalidomide. The final classifier finds a subgroup containing 19.8% of the patients that have a twofold progression free survival (PFS) benefit when treated with bortezomib and a threefold PFS benefit for lenalidomide for 31.1% of the patients. Our results demonstrate that GESTURE can be used to robustly derive clinically actionable gene expression signatures that enable a more personalized approach to cancer treatment.

## Results

**Definition of treatment benefit class.** We combined data from three randomized phase III clinical trials comprising of 910 patients with MM (see Methods), who either received the proteasome inhibitor bortezomib ($n = 407$) or not ($n = 503$). For each patient gene expression profiles were generated from purified myeloma plasma cells at diagnosis. An overall HR of 0.74 (95% CI: 0.61–0.90, $p = 0.0029$, $n = 910$) is observed between the two treatment arms, in favor of the bortezomib arm. While this HR indicates significant treatment benefit for bortezomib, we asked whether this was driven by a small benefit for all patients, or if a subgroup of patients can be identified showing a large benefit from treatment with bortezomib, while the remainder of patients show a smaller or no benefit from bortezomib. With this research we aim to identify a subset of patients, the "benefit" class, who benefit from the treatment of interest (bortezomib) relative to a comparator treatment arm which does not contain bortezomib. The patients not included in the 'benefit' class belong to the class "no benefit" and would not benefit from receiving bortezomib. The classifier identifying this "benefit" class could serve as a valuable diagnostic to determine, which newly diagnosed patients would benefit from bortezomib (based) treatment.

**Regular classifiers cannot predict treatment benefit.** We first aimed to evaluate how well a regular (prognostic) classification approach is able to reach treatment benefit prediction. According to our definition of treatment benefit, a classifier should identify a subset of patients (class "benefit") with a significantly better survival on the treatment of interest than the population as a whole. In a regular binary classification setting, training such classifier requires a labeled dataset, where the label indicates if the patient will or will not benefit from treatment. As discussed in the introduction, such labels are not available, since we cannot know how a patient would have responded to a different treatment. However, one reasonable assumption could be that patients who survive long in the treatment arm of interest do so because they benefited from the treatment, and, conversely, patients who survive short in the other treatment arm do so because they should have received the treatment of interest. Following this line of reasoning, we define the "benefit" class as the 25% longest surviving patients in the bortezomib arm and the 25% shortest surviving nonbortezomib patients. Together, these two groups form the class "benefit" (25% of all patients). All other patients from the two arms (75%) are labeled as class "no benefit".

Table 1 demonstrates that with some classifiers class "benefit" can be predicted from the gene expression data reasonably well, with a cross-validation accuracy ranging from 0.58 for the random forest classifier to 0.81 for the support vector machine classifier. However, using an independent validation fold, we find that prediction of treatment benefit fails as no improvement in HR is found over the whole population. A similar absence of performance is observed when other percentages than 25% were chosen to define the class "benefit" (Supplementary Tables 2–4).

The approach to derive labels directly from survival information is essentially similar to prognostic classification, and our results thus cast doubt on the utility of prognostic approaches in a

**Table 1 Classification accuracy in cross-validation and HR in independent validation for the classifiers trained on labels based on the top 25% surviving bortezomib patients and the bottom 25% nonbortezomib patients**

|  | Classification accuracy | Validation HR | *p* value |
|---|---|---|---|
| **Nearest mean** | 0.58 (std. dev.: 0.07) | 0.96 (95% CI: 0.57–1.60) | 0.86 |
| **Random forest** | 0.68 (std. dev.: 0.03) | 0.95 (95% CI: 0.54–1.68) | 0.87 |
| **SVM** | 0.81 (std. dev.: 0.06) | 0.81 (95% CI: 0.31–2.13) | 0.67 |

**Table 2 Classification accuracy in cross-validation and HR in independent validation for the classifiers trained on labels selected from randomly generated classifications with a significant HR under 0.5**

|  | Classification accuracy | Validation HR | p value |
|---|---|---|---|
| **Nearest mean** | 0.50 (std. dev.: 0.02) | 0.81 (95% CI: 0.49–1.35) | 0.42 |
| **Random forest** | 0.66 (std. dev.: 0.02) | 0.81 (95% CI: 0.50–1.41) | 0.51 |
| **SVM** | 0.83 (std. dev.: 0.06) | 1.10 (95% CI: 0.52–2.34) | 0.80 |

predictive setting. However, this lack of performance may not be surprising, since the training labels already lead to unrealistically large HRs ($<0.1$), indicating that the labels are often wrong. Classifiers trained on such noisy labels are indeed unlikely to have predictive performance in independent validation data. It should moreover be noted that this approach does not take censoring of the patients into account.

As an alternative approach, we therefore also generated a large number (1000) of random labelings and evaluated the HR in the "benefit" class of these randomly labeled datasets. Those labelings that resulted in a significant ($p < 0.05$) HR below 0.5 were subsequently used to train a classifier. This greedy random search procedure enables taking into account censoring of patients (through the calculation of the HR) and leads to less extreme HRs in the training data. However, this approach also did not yield classifiers with a significant HR when applied to the validation fold (Table 2). This demonstrates that it is not straightforward to derive labels for treatment benefit that can be accurately predicted from the gene expression dataset.

**Overview of simulated treatment learning**. The key idea of STL is that a patient's treatment benefit can be estimated by comparing its survival to a set of genetically similar patients that received the comparator treatment (Fig. 1d, step 1). Patients with a large survival difference compared to genetically similar patients can then act as prototype patients; new patients with a similar gene expression profile are expected to also benefit from receiving the treatment of interest. Since similarity in gene expression profile is greatly influenced by the choice of input genes, we define this similarity according to a large number of gene sets. Training the prototype-based classifier requires optimizing two parameters per gene set: the number of prototypes to use and the decision boundary, defined in terms of the Euclidean distance to the prototype (Fig. 1d, step 2). The STL classifier also needs to select the optimal gene sets to ultimately classify a patient. Importantly, the labels are now defined using the prototypes identified for the various gene sets, which means that in the STL approach there is no need to define labels before training the classifier. To train the classifier and select the best performing gene sets, the training data are split in three folds (A–C). Fold A is used to identify prototypes, fold B to optimize the decision boundary, and fold C to estimate classifier performance.

To obtain unbiased estimates of the overall prediction performance, the entire dataset is divided in three equal folds, D1–D3, ensuring a similar HR between the treatment arms in all three folds. Training is performed on two folds, while the remaining fold is kept separate to serve as an independent validation set. This is rotated to obtain an unbiased prediction for each fold. The division of the data in D1–D3, and subsequently in folds A–C is shown in Fig. 1c.

It is a priori unknown which genes will be relevant to defining patient similarity and predicting treatment response. We used 10,581 functionally coherent gene sets based on Gene Ontology (GO) annotation. Each gene set is used to train a separate classifier. The top-performing classifiers are subsequently combined into an ensemble classifier to determine the optimal number of gene sets to be used in the final classifier (Fig. 1d, step 3, for details see Methods). For the gene sets included in this optimal number a single classifier is trained using all the training data.

These classifiers are combined into the final ensemble classifier that is used to classify the patients in the validation set (Fig. 1d, step 4).

**STL finds a predictive classifier for bortezomib benefit**. Figure 2a shows the cumulative progression free survival curves for two treatment arms, with an HR of 0.74 (95% CI: 0.61–0.90, $p = 0.0029$, $n = 910$) between the treatment arms. Figure 2b shows the treatment arms and classes as identified by the STL classifier, when combining the class 'benefit' from the three validation folds. These three validation folds together comprise the whole dataset; the classification of each validation fold is predicted by separately trained classifiers. This enables us to show a validation performance for the whole dataset.

The validation HRs for the "benefit" and "no benefit" class are 0.50 (95% CI: 0.32–0.76, $p = 0.0012$, $n = 180$) and 0.78 (95% CI: 0.63–0.98, $p = 0.03$, $n = 730$), respectively. In the entire population an HR of 0.74 ($p = 0.0029$, $n = 910$) is observed. These results show that a subgroup, comprising 19.8% of the population ($n = 180$ out of 910), is identified by our method that benefits substantially more from bortezomib treatment than the population as a whole. More importantly, the STL approach is able to discover and predict this subgroup using the gene expression data at diagnosis.

In the bortezomib arm, the "benefit" and "no benefit" class exhibit similar survival curves. This is expected, since our classifier is trained to predict benefit with respect to the patient group not receiving bortezomib. As the Kaplan–Meier in Fig. 2b shows, the other treatment arm in the "no benefit" class also has a similar survival, which means we expect these patients would have had a similar survival had they not received bortezomib. The ability to determine that a patient would not benefit from bortezomib is of equal importance as predicting benefit; preventing unnecessary treatment is an important aim of personalized medicine.

The HRs observed within each of the individual validation folds are similar to the HR obtained when combining all folds (0.51 (95% CI: 0.28–0.92, $p = 0.03$, $n = 89$), 0.39 (95% CI: 0.14–1.08, $p = 0.07$, $n = 30$), and 0.46 (95% CI: 0.21–1.02, $p = 0.06$, $n = 61$) in folds D1–D3, respectively). We note that the HR is comparable in all folds, demonstrating a stable performance, although not statistically significant for fold D2 and D3 at $p < 0.05$ due to the fact that in D2 9.9% of patients and in D3 20.1% are included in the "benefit" class vs. 29.4% in D1.

Traditionally, the performance of a classifier is assessed by computing its accuracy, which is done by comparing the labels predicted by the classifier with ground truth labels. Ground truth labels are labels that are known to be accurate because they can be directly observed, e.g., if a patient survives longer than 5 years or not. Since we do not know beforehand which patients benefited from bortezomib, we have no ground truth labels available and cannot compute the accuracy of our classifier. However, we can

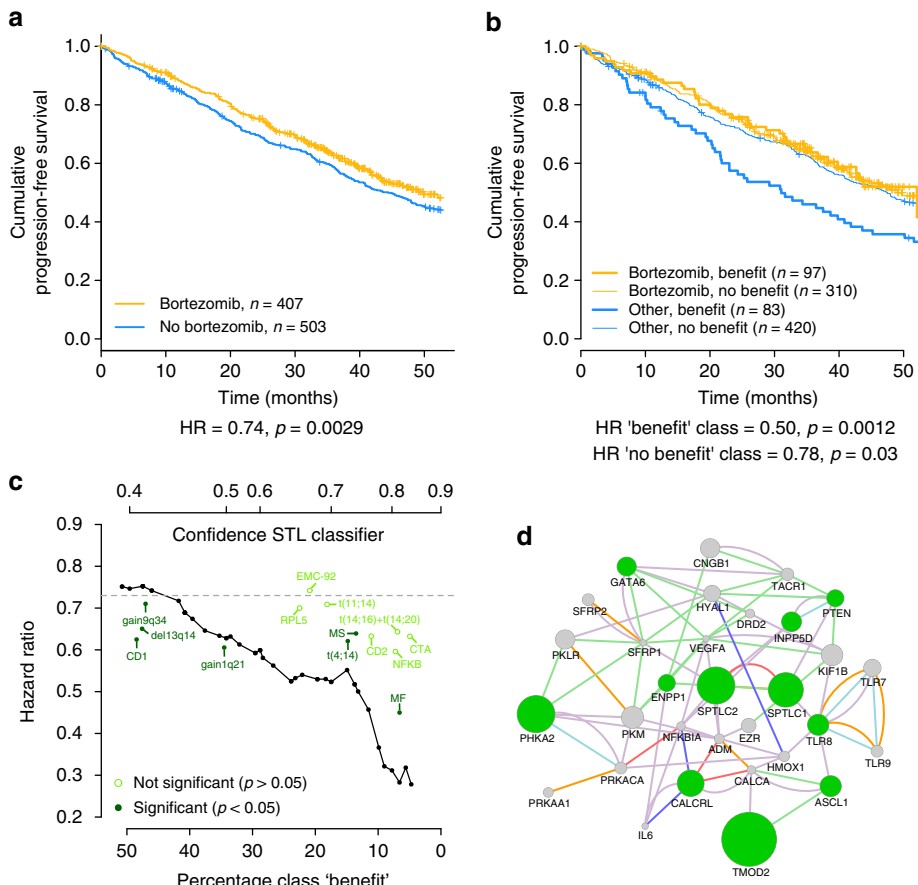

**Fig. 2** Overview of the bortezomib classifier results and comparison to known markers. **a** Kaplan–Meier of the entire bortezomib dataset, showing a HR of 0.74 (95% CI: 0.61–0.90, $p = 0.0029$, $n = 910$), between the treatment arms. **b** Kaplan–Meier of the combined classifications into a "benefit" and "no benefit" class of D1–D3. A HR of 0.50 (95% CI: 0.32–0.76, $p = 0.0012$, $n = 180$) is found between the treatment arms in the "benefit" class and a HR of 0.78 (95% CI: 0.63–0.98, $p = 0.03$, $n = 730$) in the "no benefit" class. These results show that a subgroup, comprising 19.8% of the population ($n = 180$ out of 910 total), is identified by our method that benefits substantially more from bortezomib treatment than the population as a whole; in the entire population an HR of 0.74 (95% CI: 0.61–0.90, $p = 0.0029$, $n = 910$) is found. **c** The HR found in the "benefit" class (y-axis) when different operating points (x-axis) are used, compared with known predictive and prognostic markers. The gray dotted line indicated the HR found in the entire dataset, without classification. **d** Relationships between the 31 genes in common between the D1–D3 classifiers. Node size corresponds to how much more a gene was observed in the selected gene sets than expected. Green nodes indicate that the gene is associated with a $p < 0.05$. Relationships are inferred from literature with the GeneMANIA[41] algorithm. A purple edge indicates the genes are co-expressed, a green edge indicates a genetic interaction, a red edge a physical interaction, an orange edge a shared protein domain, a dark blue edge indicates colocalization and a light blue edge shows that both genes are annotated to the same pathway

compare the class labels obtained with the three separate classifiers when applied to all 910 patients. We find that these three class assignments agree between the classifiers significantly more than expected by chance (i.e., 0/3 classifiers or 3/3 classifiers predict benefit; Supplementary Figure 1). A similar conclusion is reached by comparing the classification scores directly, which significantly correlate (all $p$ values $< 1 \times 10^{-4}$). When considering the cases for which the 3 classifiers agree, we find that 503 patients are consistently classified as "no benefit" and 57 patients as "benefit". Together, this demonstrates that, even though the classifiers do not agree on the class assignment for all patients (which is expected in practice for classifiers with less than 100% accuracy), they capture the same gene expression patterns.

The decision boundary of the classifiers are defined by the parameters k and γ and a threshold $T$. We optimize the combination of k and γ by an exhaustive grid search. We verified that the performance of our classifier is robust to small changes in these parameters (Supplementary Note 1).

The operating point of the classier is determined by the number of individual classifiers in the ensemble that agree on the class label, and is thus directly related to the confidence of the ensemble classifier about the label "benefit". To ensure sufficient power and provide a treatment decision for a substantial group of patients, the operating point of the classifier was set to 20% in training (see methods). At this operating point, 19.8% of patients in the validation folds were actually assigned to the 'benefit' class. Figure 2c depicts the HR as a function of the confidence level of the classifier. We observe that, for higher confidence levels (yielding smaller sizes of the "benefit" class) more extreme validation HRs are observed, demonstrating that there is a direct relation between classifier score and treatment benefit. This is consistent with the fact that the highest HR and largest class "benefit" are found in fold D1 in validation, while the lowest HR and the smallest class "benefit" are found in D2.

As a control experiment, we also ran the algorithm with shuffled treatment labels, destroying the relationship between the gene expression and the treatment specific survival. As expected,

the classifier trained on this data shows no performance in the validation data, achieving an HR of 1.09 (95% CI: 0.71–1.67, $p = 0.69$, $n = 167$) in the class "benefit" and an HR of 0.95 (95% CI: 0.77–1.18, $p = 0.65$, $n = 743$) in the class "no benefit" (Supplementary Figure 3). This reinforces our observation that STL identifies a true effect, since the classifier shows no performance in random data.

**STL classifier outperforms known markers**. We compared the HRs found using the STL classifier with several known prognostic markers in MM, some of which also show predictive value (Fig. 2c). The STL classifier has a superior performance for operating points that result in assignment of up to 30% of the patients to the class "benefit". The markers that slightly outperform the STL classifier do so only for operating points that results in much larger sizes of the class "benefit" and lead to smaller effect sizes. The gray line indicates the baseline HR found in the entire dataset. A clinically actionable classifier should reach a substantially larger benefit than this baseline, which is only attained by the STL classifier and the MF cluster for operating points <30%, where the STL classifier outperforms the MF biomarker.

**Biological information is important for performance**. To investigate if the biological knowledge contained in the GO, used to define gene sets, truly aids classification performance, we also tested random gene sets with the same set size distribution. Using the random gene sets, final classification results in a significant HR of 0.56 (95% CI: 0.34–0.90, $p = 0.02$, $n = 148$) when all three validation folds are combined (Supplementary Figure 2).

This is not unexpected as combining random feature sets in an ensemble classifier is known to achieve good classification performance[19]. Moreover, it has been shown previously that random gene signatures can perform on par in a prognostic setting[20]. Nonetheless, the STL classifier trained using the GO gene sets outperforms the random gene set approach in both HR and $p$ value. Moreover, in contrast to the relatively stable performance across validation folds when using the GO gene sets, the performance of the random set approach varies greatly between the folds, ranging from an HR of 0.76 (95% CI: 0.32–1.85, $p = 0.55$, $n = 41$) in D1 to an HR of 0.44 (95% CI: 0.21–0.93, $p = 0.03$, $n = 67$) in D3.

Together, this demonstrates that the biological information contained in the GO gene sets is important to the performance of the STL classifier.

**Genes used to predict treatment benefit bortezomib**. The classifiers built for D1–D3 use 113, 218, and 111 GO gene sets, respectively to predict bortezomib benefit, encompassing a total of 1913 unique genes. There are 31 genes used in all three classifiers (Fig. 2d). There are GO categories that include a large subset of these 31 genes, including "positive regulation of transcription from RNA polymerase II promoter", "cellular response to hypoxia", and "negative regulation of the apoptotic process". All these GO categories are associated with the pathogenesis of cancer. Both increased proliferation and the ability to evade apoptosis are hallmarks of cancer[21]. It has also been established that cancer cells can adapt their metabolism to thrive in hypoxic conditions[22]. For the 31 genes, we calculated whether they are selected more than expected by chance. GO sets are hierarchical (i.e., there is a larger parent category that can include several children categories) and genes can be annotated to multiple GO categories. Therefore, we have taken into account how many GO categories include a certain gene to establish if we observe a gene more often than expected in our classifiers. The expected count

for a gene is based on the number of GO categories that include that gene, e.g., *PTEN* is included in 123 of the 10,581 gene sets, so in the 442 gene sets used across D1–D3 we would expect to observe *PTEN* approximately 5 times if it would occur at the same frequency as within our selected gene sets. Most genes in common between the three classifiers are observed more often than expected (degree of overrepresentation indicated by node size in Fig. 2d), with 11 of 31 significantly overrepresented ($p < 0.05$). The most overrepresented genes are *TMOD2*, *PHKA2*, *SPTCL1*, and *SPTCL2*. None of these genes are known to be associated with MM or response to bortezomib. However, investigation of the proteome of a cell line carrying a *SPTCL1* mutation showed an increased presence of Ig kappa chain C[23]. Immunoglobulin light chain presence is used as a biomarker for MM and has been identified as a risk factor for progression[24]. *PTEN* is also found to be significantly overrepresented. *PTEN* is a known tumor suppressor and was found to be mutated in a various cancers[25]. In MM, *PTEN* mutations are relatively uncommon and associated with advanced disease[26].

**Impact of dataset of origin on validation performance**. Our training dataset is a combination of three different datasets: total therapy 2 (TT2), total therapy 3 (TT3) (together forming the TT dataset), and HOVON-65/GMMG-HD4 (H65). Both the bortezomib and the no bortezomib arm contain more than one treatment regimen (Supplementary Table 1). We trained and validated on a combination of the datasets (see Methods). To investigate the contribution of the different datasets to the final validation performance, we calculated the HR in class "benefit" for the TT and H65 patients separately. Reassuringly, we observe a similar effect in class "benefit" in both datasets, albeit not significant due to small sample size in the H65 dataset (HR = 0.69 (95% CI: 0.36–1.32), $p = 0.26$, $n = 49$, for H65 and HR = 0.38 (95% CI: 0.21–0.69), $p = 0.002$, $n = 131$ for TT, Supplementary Figures 4, 5). Also, the observed HR is much smaller in the TT dataset. This may be expected, since the HR in the overall population is also smaller in TT than in H65 (the overall HR in TT is 0.62 (95% CI: 0.46–0.84), $p = 0.002$, $n = 583$ vs. an HR of 0.86 (95% CI: 0.66–1.13), $p = 0.28$, $n = 327$ in H65).

We hypothesized that heterogeneity helps to prevent overfitting to one specific dataset or treatment regimen. To test this, we also performed a cross-validation within the two TT datasets only (the H65 dataset is too small for this with $n = 327$). Subsequently, we trained a classifier on the entire TT dataset (combining TT2 and TT3) and validated on H65. Cross-validation within the TT dataset leads to an HR of 0.28 (95% CI: 0.13–0.60, $p = 0.00098$, $n = 86$) in class "benefit" and an HR of 0.71 (95% CI: 0.51–0.98, $p = 0.038$, $n = 497$) in class "no benefit" (Supplementary Fig. 6), which is a substantial improvement over the classifier trained on the combined dataset. In contrast, when the classifier is trained on the entire TT dataset, no performance is observed in the H65 dataset (an HR of 1.13 (95% CI: 0.63–2.04), $p = 0.68$, $n = 66$ in class "benefit" and 0.81 (95% CI: 0.60–1.1), $p = 0.18$, $n = 261$ in class "no benefit"), indicating that some dataset specific fitting has occurred. Importantly, dataset specific fitting does not necessarily indicate overtraining; the classifiers still validate on the completely independent hold out validation fold. These results do suggest that it is very important to match the training population with the population one intends to use the classifier in. If the population in which the classifier is intended to be applied is heterogeneous, the training dataset also needs to reflect this heterogeneity.

In the MM dataset under study here, one possible explanation for the lack of validation of the TT-based classifier on the H65 data is that the TT trials were conducted in the USA and included

more additional treatment than the European H65 trial (see Supplementary Table 1 for treatment details). When the STL classifier is trained exclusively on the TT datasets, it could become specifically predictive for the TT regimen, rather than bortezomib, explaining why this classifier does not show a satisfactory performance in H65. When trained on the mixed dataset, the classifier does show performance in the H65 dataset, but still performs better within the TT dataset, which makes up a bigger part of the training data.

**STL finds a predictive classifier for lenalidomide benefit.** The STL method was developed based on the bortezomib dataset. Even though a strict separation of training and validation has been made, we cannot exclude the possibility of "experimenter bias"[27], which is the result of making experimental choices based on the results on the training dataset and which can lead to a classifier that will only perform well on the specific dataset at hand.

To demonstrate that the STL method is not biased to just one dataset we applied it to a completely independent dataset obtained from the CoMMpass database (https://research.themmrf.org/). CoMMpass contains data from an observational MM study, meaning the trial did not interfere with the treating physician's choice of treatment. This is a good model for the setting in which an eventual predictive biomarker would be applied. Moreover, instead of microarrays, RNAseq was used to obtain gene expression measurements, thus providing an additional axis of variation compared to microarray data. Overall, gene expression data and annotation was available for 662 patients, 447 of which received lenalidomide in the first line and 215 did not. An overall HR of 0.59 ($p = 0.004$) in favor of lenalidomide was observed, as seen in the Kaplan–Meier in Fig. 3a.

Similar as before, the dataset was divided into three equal folds and STL obtains classifiers that successfully predict benefit in all folds. Since the CoMMpass dataset is smaller than the bortezomib dataset used before, we required the "benefit" class to contain at least 30% of the patients, to ensure sufficient power. This results in a combined HR of 0.36 (95% CI: 0.18–0.71, $p = 0.0031$, $n = 206$) over the entire dataset, as shown in Fig. 3b. In total 31.1% of patients were classified as class "benefit". Again, the STL classifier was able to distinguish a subset of patients with significant treatment benefit in each fold with HRs of 0.27 (95% CI:

0.07–1.06, $p = 0.06$, $n = 72$), 0.39 (95% CI: 0.11–1.41, $p = 0.15$, $n = 66$) and 0.40 (0.14–1.15, $p = 0.09$, $n = 68$) in D1–D3, respectively. This demonstrates that STL also successfully identified a predictor for lenalidomide benefit.

**Genes used to predict treatment benefit lenalidomide.** The predictive classifiers for lenalidomide use 47, 5, and 119 gene sets in D1–D3, respectively, encompassing 3723 unique genes. Out of these, 5 genes are used in all three classifiers: *CYP11B2*, *SHH*, *HGNC*, *CAV1*, and *SMO*, all of which are observed more frequently than expected. *SHH* and *CYP11B2* are significantly overrepresented ($p < 0.05$). *SHH* is a crucial part of the hedgehog signaling pathway, which has been previously found to play an important role in the pathogenesis of MM[28]. Neither of these genes has previously been associated with lenalidomide response, possibly representing an undiscovered mechanism influencing lenalidomide response in MM patients.

## Discussion

Simulated treatment learning addresses an urgent clinical need because response rates to current cancer therapies are often poor and moreover frequently accompanied with serious side effects. STL offers an important step towards realistic personalization of cancer medicine administration by identifying gene expression markers that can be used to determine the most effective treatment for a cancer patient at the moment of diagnosis.

The STL classifier was successfully tested across different gene expression platforms, different treatments and different study types, demonstrating that STL is more generically applicable than one particular dataset. Since our work has focused on MM, an important next step is to investigate if STL is also successful in unraveling treatment benefit for other diseases. If so, STL can play an important role in rescuing treatments that do not achieve a significant effect in the entire patient population but may still benefit a subset of the patients. For instance, STL can be an important post hoc analysis for phase III clinical trials of novel treatments that have missed their endpoint, such as, for instance, nivolumab in the CheckMate-026 trial[29]. We do note that STL requires a relatively large number of samples to build the classifier, which may not always be available when a novel treatment first enters clinical trials. The generic concept of STL can be readily extended to include patient similarity definitions based on

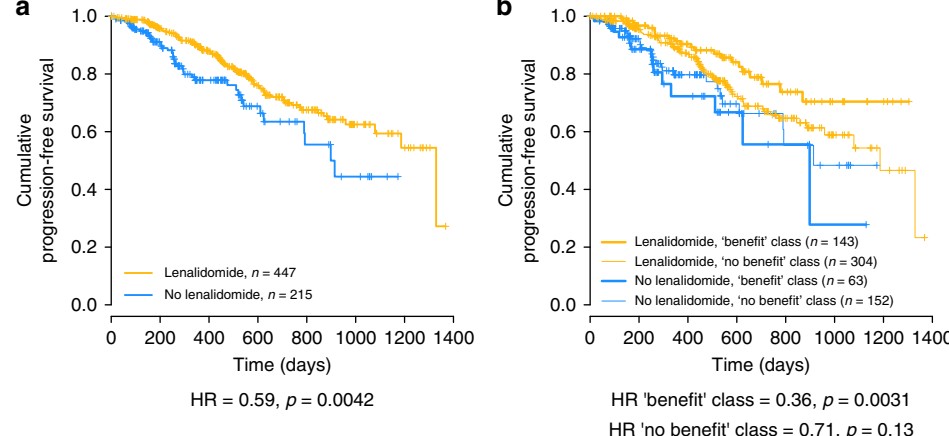

**Fig. 3** Overview of the lenalidomide classifier results. **a** Kaplan–Meier curves for the entire lenalidomide dataset, showing an HR of 0.59 (95% CI: 0.41-0.84, $p = 0.0042$, $n = 662$) between the treatment arms. **b** Kaplan–Meier curve of the combined classifications into a "benefit" and "no benefit" class of D1–D3. An HR of 0.36 (95% CI: 0.18–0.71, $p = 0.0031$, $n = 206$) is found between the treatment arms in the "benefit" class and an HR of 0.71 (95% CI: 0.46–1.10, $p = 0.13$, $n = 456$) in the "no benefit" class

e.g., germline or somatic genomic profiles and other types of outcome measure such as categorical or binary measures.

## Methods

**Data and processing.** We pooled gene expression and survival data from three phase III trials: TT2 (GSE2658), TT3 (GSE2658), and H65/GMMG-HD4 (GSE19784). The TT2 dataset included 345 newly diagnosed MM (NDMM) samples, treated either with thalidomide and melphalan ($n = 173$) or melphalan alone ($n = 172$). Average age is 56.3 (range: 24–76) and 57.1% of the patients is male. The TT3 dataset included 238 NDMM samples treated with bortezomib, thalidomide, dexamethasone, cyclophosphamide, cisplatin, and etoposide. Average age is 58.7 (range: 32–75) and 67.6% is male. The H65 dataset included 327 NDMM samples, treated either with vincristine, doxorubicin, and dexamethasone (VAD, $n = 158$) or bortezomib, doxorubicin and dexamethasone (PAD, $n = 169$). Average age is 54.7 (range: 27–65) and 56.4% is male. In our analyses of the pooled data two treatment arms were considered: a bortezomib arm, which comprises the PAD arm from H65 and TT3, and a nonbortezomib arm, which comprises the VAD arm from H65 and TT2. Combined, these datasets include 910 patients, of which 407 received bortezomib and 503 did not.

All samples were profiled with the Affymetrix Human Genome U133 plus 2.0 array. Gene expression was MAS5 and log2 normalized. Batch effects resulting from pooling different datasets were corrected with ComBat[30]. Data were scaled to mean 0 and variance 1 per probeset. Probesets with a variance of <1 before scaling were discarded.

The data was split in fold D1 (303 samples), fold D2 (303 samples), and fold D3 (304 samples), stratifying for treatment arm and survival. Fold D1 is not used at any point in the training and serves as validation data, while fold D2 and fold D3 are combined to serve as training data. After the STL classifier is successfully validated on fold D1, the folds are rotated to serve as additional validation folds to assess robustness. The training data for fold D2 consists of D1 and D3 and the training data for fold D3 consists of D1 and D2 (specification of which samples were used in which folds is available with the code in the GitHub repository).

After developing the STL method on the microarray dataset, we also applied it to the CoMMpass trial (NCT0145429) dataset generated by the Multiple Myeloma Research Foundation. For 662 patients both RNAseq, survival data, and treatment information was available. Sequencing data is processed with the Cufflinks pipeline (researcher.themmrf.org). The dataset was split into a treatment arm where patients received lenalidomide as first-line treatment ($n = 447$) and an arm where patients did not ($n = 215$). This data was also split into folds D1 (220 samples), D2 (221 samples) and D3 (221 samples), specification of which samples were used in which folds is available with the code in the GitHub repository.

**Endpoint and survival analysis.** PFS was used as endpoint, as this is the most direct readout of first-line treatment related survival, and therefore considered to be more relevant compared to overall survival. PFS times in the TT2 and H65 datasets were truncated to 52.53 months, corresponding to the longest follow-up time in the TT3 dataset.

Survival analyses were done using the Cox Proportional Hazards model (survival package, version 2.38.4)[31]. For the microarray data, the survival analysis included a stratification for dataset of origin. This means the base hazard was estimated separately for the TT2/TT3 dataset and the H65 dataset. This is necessary to correct for the significant survival difference found between these datasets. HR and associated two-sided $p$ values were calculated. $P$ values below 0.05 were considered statistically significant. All HRs are computed as bortezomib vs. no bortezomib and lenalidomide vs. no lenalidomide, which means an HR below 1 signifies a benefit when receiving bortezomib or lenalidomide. All calculations were performed in R version 3.1.2.

**Gene sets.** For the bortezomib classifier we tested all GO categories, as defined by the R Bioconductor package hgu133plus2.db[32] (accessed 27 October 2015), with two or more probesets associated to them. This resulted in 10,581 gene sets. To test whether the biological information, contained in the GO annotation, aids the performance of the algorithm, 10,581 random gene sets matching the size of the actual selected GO categories were also tested.

For the lenalidomide classifier we tested all the GO categories with two or more genes associated to them, as defined by Bioconductor package biomaRt[33] (accessed 19 June 2017). This resulted in 9121 gene sets.

**Algorithm.** The STL classifier aims to predict if a patient does or does not benefit from a certain treatment of interest based on the gene expression profile of the patient. In order to train this classifier, a gene expression dataset is required that consists of two treatment arms and a continuous outcome measure. These data are first split into training and validation folds. The training data comprises of two-thirds of the data, while one-third (fold D) is kept apart to function as validation data. We define three separate folds D (D1–D3), such that each patient is included in the validation set once. The training data is subsequently split further into folds A–C for training.

We first define a ranked list of prototype patients on fold A (Step 1) that exhibit a better than expected prognosis on the treatment of interest compared to a set of genetically similar patients that received an alternative treatment. In Step 2, a decision boundary around a selection of prototype patients is determined on fold B. Patients that lie within this decision boundary are expected to show a favorable outcome when receiving the treatment of interest and are classified as benefitting (class "benefit"). All other patients are considered class "no benefit" and are not expected to benefit from receiving the treatment of interest. Because it is a priori unknown based on which genes patient similarity should be defined, step 1 and 2 are performed for a large number of functionally coherent gene sets obtained from the GO annotation, yielding one classifier per gene set. Step 1 and 2 are repeated 12 times to obtain a robust estimate of the performance per gene set. In each repeat, the training data is split into a different fold A–C. The performance is defined as the HR between treatments in class "benefit", found in a fold C, which contains samples that were not used in step 1 and 2. All gene sets are ranked by their mean performance in fold C across repeats. In Step 3 we determine the optimal number of gene sets to combine into a final classifier. We found that defining performance and selecting the optimal number of gene sets on the same folds C leads to overtraining. Therefore, we run the entire algorithm a second time (Run 2), using 12 new repeats with different splits into fold A–C. The first run of 12 repeats is used to rank the gene sets. The combined performance of these ranked gene sets on the folds C from Run 2 is used to determine the optimal number $s$ of gene sets. Similar to the boosting principle[34], the individual classifiers are combined into an ensemble to construct a more robust final classifier. The performance of this combined classifier is measured on fold C of Run 2. The gene sets are added to the classifier in order of their ranking, until an optimal performance is reached across all the repeats from Run 2. Since there are 12 repeats, each combination results in 12 HRs as measured on the folds C from run 12. To determine the optimal number of gene sets, we fit a local polynomial regression line on the median HRs for each combination of gene sets. The optimal number of gene sets $s$ is reached when adding a gene set does not result in a lower HR. We then rank the gene sets based on their individual performance across the folds C of Run 2 and select the top $s$ for inclusion in the final ensemble classifier. Finally, in Step 4, one final classifier is trained using the entire training dataset for these selected gene sets.

These steps are visualized in Fig. 1d and are described in more detail below.

In Step 1, we perform prototype ranking on fold A. For each patient receiving the treatment of interest, the treatment benefit is defined as

$$\Delta\text{PFS}_i = \frac{1}{n}\sum_{j\in O}\left(\text{PFS}_i - \text{PFS}_j\right) \tag{1}$$

where $O$ is the set of the $n$ most similar patients (based on Euclidean distance) that did not receive the treatment of interest. We use $n = 10$. In an approach similar to Harrell's C-statistic[35], $\Delta\text{PFS}$ is only calculated for neighbor pairs where it is clear which patient experienced an event first; if both are censored or one patient is censored before the neighbor experienced an event, $\Delta\text{PFS}$ is not computed. When $n = 10$ is used, this on average leads to seven neighbors being used in the calculation of $\Delta\text{PFS}$. To correct for the fact that a patient with a long survival time will, on average, have a large $\Delta\text{PFS}$ irrespective of its relative treatment benefit compared to genetically similar patients, we define the z- normalized $z\text{PFS}$ score as:

$$z\text{PFS}_i = \frac{\Delta\text{PFS}_i - \mu(\text{RPFS}_i)}{\sigma(\text{RPFS}_i)} \tag{2}$$

where RPFS is a distribution of 1000 random $\Delta\text{PFS}$ scores, obtained by calculating $\Delta\text{PFS}$ for randomly chosen sets $O$, i.e., determining treatment benefit with respect to random patients instead of genetically similar patients. Based on the $z\text{PFS}$ score all patients in fold A that were given the treatment of interest can be ranked.

In Step 2, we define the classifier on fold B. The classifier is defined by a subset of $k$ top-ranked prototypes along with a decision boundary defined in terms of the Euclidean distance $\gamma$ around a prototype. A patient is classified as class "benefit" when it lies within $\gamma$ of any of the top $k$ prototypes. The optimal values for $k$ and $\gamma$ are those resulting in the lowest HR in class "benefit" (the patient group in which the treatment of interest should have a better survival). We set an operating point that additionally constrains $k$ and $\gamma$, such that class "benefit" comprises at least a certain percentage of the dataset. This ensures sufficient statistical power to compute the significance of the HR in the "benefit" class. The number of prototypes was restricted to ten to prevent defining an extremely complicated classifier. The search grid for parameter $\gamma$ was made dependent on the local density of the neighbors, and consisted of the sorted list of Euclidean distances between the prototype and its neighbors. The optimal $k$ and $\gamma$ combination is chosen so that the HR in class "benefit" is minimal, while still associated with a $p$ value below 0.05. If no combination results in a $p$ value below 0.05, the minimal nonsignificant HR that results in a class "benefit" of sufficient size is chosen.

In step 3, we rank and select the gene sets. First, the gene sets are ranked by their mean performance in fold C over all repeats from Run 1. After ranking, we run the algorithm a second time, with different divisions into fold A–C. We add gene sets to an ensemble classifier one by one based on this ranking. The performance of the combined gene sets is measured on each fold C of this second run. We find that defining the ranking on different folds than we use to measure combined performance prevents overtraining, although some bias is still expected

to occur. Since the found HR can fluctuate between folds and gene set numbers, a regression line is fit through the median HRs found on folds C in the second run and the optimal number of gene sets is determined: the first combination of gene sets for which adding another gene set does not lead to an improvement of the HR larger than $1 \times 10^{-4}$.

After the optimal number of gene sets is determined in Step 3, the final classifier is defined in Step 4. The gene sets are ranked based on their mean performance in fold C in the second run. The top scoring gene sets are selected and for these gene sets a final classifier is trained. To this end, the complete training dataset is split into only two folds, since the third fold is no longer required. The classifiers defined by different gene sets are combined into an ensemble classifier by an equally weighted voting procedure, which means each classifier has an equal influence on the final classification. For an ensemble classifier containing $s$ gene sets, this defines a classification score between 0 and $s$ per patient. This score is thresholded by threshold $T$, which determines whether a patient is to benefit from the treatment of interest, where a patient with a score below the threshold is classified as not benefitting from treatment ("no benefit" class). The optimal threshold $T$ is the one for which the HR between treatments is minimal in class "benefit". This combination of classifiers and threshold can be used to classify new and unseen patients and is validated on fold D.

**Calculating overrepresentation of genes in the classifier**. The same gene can be used multiple times in a single classifier and/or multiple times across the classifiers obtained for fold D1–D3. Both cases provide evidence of the importance of the gene for the treatment benefit prediction. To assess whether genes are selected more frequently than expected by chance across all three classifiers, we determine the degree of overrepresentation by dividing the observed count by the expected count. The expected count is calculated by $p \times W$, where $p$ is the fraction of the gene sets containing the gene and $W$ the total number of gene sets selected across all three classifiers. A $p$ value is determined using the binomial test.

**Training regular classifiers**. We defined the labels that were used to train the regular classifiers in two ways. First, labels were defined by assigning the 25% longest surviving bortezomib patients and the 25% shortest surviving non-bortezomib patients to the "benefit" class and all others to the "no benefit" class. A classifier was trained using folds A–C to predict these labels, using the HR in validation fold D1 as performance measure of the predictive power. For the nearest mean classifier, a double-loop cross-validation was used to optimize the number of genes (ranked based on $t$-score), using balanced accuracy as the performance measure.

A random forest classifier (R package randomForest, version 4.6.12)[36] and a support vector machine (R package e1071, version 1.6.7)[37] were also trained. For both these classifiers, the number of genes was optimized in cross-validation. For the random forest classifier 2000 trees were trained per classifier and the bootstrap sample was sampled equally from both classes, to prevent the classifier being affected by the class imbalance. For the support vector machine, $C$ values from 1 to 100 were tested, in steps of 1. The gamma used is $1/P$, where $P$ is the number of input variables, i.e., the number of genes.

For all classifiers, the accuracy reported is the mean accuracy in cross-validation for the optimal number of input genes.

**Comparison with known prognostic markers**. To the best of our knowledge, RPL5 is the only published gene expression-based marker that predicts bortezomib benefit by comparing to another treatment group[18]. We tested RPL5 on the data from the TT studies, since it was trained on the H65 data. Since some predictive markers are discovered by testing markers previously known to be prognostic, we also compare with prognostic markers. FISH markers were called on the gene expression data, using previously developed classifiers[38], since FISH data were not available for all patients. Unfortunately, there is no reliable gene expression classifier for del17p. We tested if any predictive information was available in previously defined molecular subtypes in MM[39] and in the prognostic gene signature EMC-92[40].

**Code availability**. All code needed to train and validate the classifier is available at github.com/jubels/GESTURE.

**Data availability**. All survival and treatment data included in the bortezomib dataset are supplied in the GitHub repository. The gene expression data from the TT2 and TT3 studies are accessible in the GEO database, accession number GSE2658. The gene expression data from the H65/GMMG-HD4 study is accessible in the GEO database, accession number GSE19784.

All survival, treatment and RNAseq data used for the lenalidomide dataset is accessible at research.themmrf.org.

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

## Acknowledgments

This work has been supported by a grant from the Van Herk Fellowship. The lenalidomide dataset was generated as part of the Multiple Myeloma Research Foundation Personalized Medicine Initiatives (https://research.themmrf.org and www.themmrf.org). We thank Rowan Kuiper for data aggregation and his advice on combining the datasets.

## Author contributions

J.d.R., M.H.v.V., and J.U. developed the algorithm. J.U. implemented the algorithm and performed data analysis. J.d.R. and J.U. wrote the manuscript. M.H.v.V, E.H.v.B, P.S., and A.B. provided comments and edits. All authors discussed the results.
