## [Peer Review File · Nature Communications]

Reviewers' comments:

Reviewer #1 (Remarks to the Author):

Simulated Treatment Learning (STL) is an interesting concept that uses genetic profiles among patient receiving different treatments but have similar genetic profiles – and using these to model each other's response to those treatments. This is followed by implementation with Gene Expression-based Simulated Treatment Using similarity between patients (GESTURE). The authors apply this to two large MM GEP data sets using Btz and Len.

The concept seemed straightforward, but the description of results became muddy for this reader. As a result, the paper needs some improved clarity in the description of data analysis, and importantly its application. There appear to be some similarities that are emphasized, but some inconsistencies that are noted are dismissed. For that reason, some of the analyses became unclear in how robust this approach is.

Some specific concerns:

1. While possibly understood by the reader, it is important to note that statement about using “patients who have genetically similar expression patterns” refers specifically to the tumor cells, not the individual.
2. While patients are separated into +Btz and no-Btz treatments, it is really important to note that combining data from Btz regimens includes additional drugs that may differ in combining effects – that are well known.
3. The key to the presentation lies in Fig 2; yet the descriptions are confusing and need clarification. From the K-M curves of Fig 2b, it appears Btz benefit and no benefit are the same. Critical to the conclusions is the statement that 19.8% (n=180) show benefits. It is not clear how this number was derived. Further, is the conclusion that the classifier is “showing” benefit or predicts or fits the benefit classifier used in the analysis?
4. The authors state “...there are no ground truth labels available...” What is a ground truth label?
5. The authors state that the classifiers capture the same gene patterns, but “they do not agree on class assignment for all patients...” Isn't this a problem?
6. The authors nicely include analysis of random data sets; but note they can reach significance, just not as good as the STL approach. Still, this seems a problem in overfitting the data used in the classifier modeling.

The concepts presented are interesting, and may provide some important new insights into better personalized directions for patient treatment. For this reader, the applications were missed because of a need for more clarity in the approaches described in the main text.

Reviewer #2 (Remarks to the Author):

The manuscript by Ubels and coworkers uses machine learning to predict the response of individual patients to alternative therapies, based on the patient's gene expression profile. The aim is to use these predictions for personalizing therapy decisions. While numerous approaches exist that successfully predict patient survival based on expression profiles, the question asked here is different: Decide if a given patient will benefit from a specific therapy, or not.

From a machine learning point of view, this is a difficult problem, as in any clinical trial it is only known how a patient has responded to the therapy he or she has actually received, but it is unknown how the patient would have responded to alternative treatment. The authors solve this problem by pooling patients with similar expression profiles, and within these pools compare survival of the subgroups of patients receiving therapy A vs. B. The underlying assumption is that similar gene expression profiles imply similar therapy response.

From a biometrical point of view, the identification of subgroups that benefit from a given therapy (versus those subgroups that do not) is a standard procedure. If a hypothesis exists which marker or combination of markers might define such subgroups, testing if there is a benefit of one treatment over the other in one of the subgroups, but not the other, is a trivial task. The main contribution of the present manuscript is to develop a systematic procedure to test different multivariate, gene-expression based "subgroup" definitions, by taking gene ontology (GO) information into account and defining patient similarity as similarity of expression profiles of gene subsets defined by GO groups.

The authors have done a good job describing their algorithm, and follow a rigorous training procedure based on crossvalidation. From a statistical point of view, there are some methodological flaws which I detail below. The approach presented is novel, and could help personalize cancer treatment. The main flaws I see concern an improper handling of censoring in the analysis that will introduce a bias, the somewhat ad-hoc procedure that requires manual setting (or grid search) for a number of parameters and the lack of an independent replication study to confirm results. I detail these and some additional points below:

- The procedure used to train the "regular" prognostic classifier (fig 1) is flawed and very ad-hoc, and it is no surprise that this is not working. The authors basically compare the top 25% longest surviving patients in the bortezomib group, and compare them against the worst 25% in the "other" treatment group. This mixes two different effects: Survival differences due to other factors, and survival differences due to treatment. It is impossible to attribute any survival differences to one of the two factors with this setup, and in fact they might even cancel each other. Furthermore, the 25% cutoff is very ad hoc, and may not be optimal. Worse yet, the comparison with their predictor is unfair, as only a fraction (50%) of the training data is used for training the "regular" classifier, versus all training data for training of their classifier. Patients that do not do well on Bortezomib and patients that do well with alternative treatment will also contribute important information, and should not be ignored. A final problem comes from the discretization of survival groups, where

patients are assigned to "good" or "bad" survival. How are censored survival times handled when identifying the top 25% survivors? If patients with early censoring are ignored, this will introduce a bias in the analysis, as censoring will more affect the "good prognosis" group than the bad prognosis patients. In the extreme case, consider a curative treatment, where all patients receiving the therapy would be censored for progression free survival. The "best" patients may thus be filtered out in the good prognosis group. And this would decrease the difference between the "good" and the "bad" group...

- The same bias due to censored patients arises in the identification of prototype patients in the first step of the presented "new" algorithm. Can the authors comment on this? How strongly will censoring bias their results?

- There are a number of parameters that need to be set in the authors' approach, and it is unclear how they should be optimally chosen and what influence the choice has on results. For example, in step one, the prototype group size is chosen as $n=10$. This parameter will likely have a significant impact, and balances bias against variance of the prototype computations. The effect of the choice of n should be systematically assessed. Parameters k and γ in step 2 and T in step 4 are chosen using grid search, can the authors comment how these parameters influence results?

- While the authors show good results in a crossvalidation setting on two multiple myeloma datasets (one focusing on Bortezomib, the other on Lenalidomide treatment), the validation of the predictive models is done on the same datasets (using a hold-out validation set). There are numerous studies that show that performance on completely independent data sets will decline, hence a further, completely independent dataset would be very useful for model assessment. The ideal situation would of course be a prospective study assessing the performance of the model in assigning treatments over a treatment decision according to guidelines. Clearly, this is not feasible; however, could other MM/Bortezomib studies be used to at least perform a validation on an independent data set? If this also is not feasible, what would happen if the authors train their predictor on the TT3 data set and make predictions on the H65 data?

- Along those lines, as the authors claim that their approach is generalizable, I feel a second disease should be chosen and similar analyses be shown.

- GO contains hierarchical gene groups. Is this taken into account in the algorithm presented? If not, the analysis of how often a gene occurs in different predictors may simply be due to a larger number of hierarchy levels for that gene, and may not give any indication of "biological importance"

- Please provide confidence intervals for computed Hazard Ratios and list the exact test used along with n for all reported p -values; clarify for all computed hazard ratio whether the ratio is computed as A/B or B/A . Furthermore, please provide p -values for all computed HRs, e.g. in Table 1

- Please provide more details on the Multiple Myeloma data sets used, in particular characteristics of patients (age, gender, immunoglobulin subtype, status of genetic markers such as 17p, etc)

- The authors explain the difference between what they call "prognostic" and "predictive" signatures in Figure 1a and b. They define "prognostic" signatures as signatures that do not take treatment into account in the text, but the example they show in Figure 1a as a "prognostic" model is a classifier that is trained to distinguish the patients with best survival in the treatment group against the 25% worst survivors in the control group. This is inconsistent with their previous definition.

- I did not understand supplementary figure 1. Could you please explain in more detail what is shown and what the conclusions are?

We have carefully considered the two reviewers' comments and were happy to read a general appreciation among the reviewers of the innovativeness of the approach and promise to deliver new insights into better personalized directions for patient treatment. We are grateful to the reviewers as their comments have prompted us to update the manuscript, improving our manuscript as a result. More specifically, we have added additional experiments, showing the robustness of our classifiers when parameters are (slightly) changed. Moreover, prompted by the reviewer comments, we have also investigated the effect of using the different MM datasets together as training and validation data.

Please find below a copy of the reviewer comments with our answers in blue.

Simulated Treatment Learning (STL) is an interesting concept that uses genetic profiles among patient receiving different treatments but have similar genetic profiles – and using these to model each other's response to those treatments. This is followed by implementation with Gene Expression-based Simulated Treatment Using similaRity between patiEnts (GESTURE). The authors apply this to two large MM GEP data sets using Btz and Len.

The concept seemed straightforward, but the description of results became muddy for this reader. As a result, the paper needs some improved clarity in the description of data analysis, and importantly its application. There appear to be some similarities that are emphasized, but some inconsistencies that are noted are dismissed. For that reason, some of the analyses became unclear in how robust this approach is.

Some specific concerns:

1. While possibly understood by the reader, it is important to note that statement about using “patients who have genetically similar expression patterns” refers specifically to the tumor cells, not the individual.

It is indeed important to note that we are comparing similarity between the tumors profiled, not the patient's own genetic profile.

Changes to the manuscript:

- We changed the sentence introducing STL to “To address this, we introduce the concept of Simulated Treatment Learning (STL), a method to derive classifiers that can predict treatment benefit. STL can be applied to gene expression datasets with two treatment arms and survival data. STL uses genetic similarity, defined based on gene expression in the tumor, between patients from different treatment groups to model how a particular patient would have responded to the alternative treatment.”

2. While patients are separated into +Btz and no-Btz treatments, it is really important to note that combining data from Btz regimens includes additional drugs that may differ in combining effects – that are well known.

The reviewer rightfully points out that there are significant differences in the bortezomib arm included in the HOVON-65/GMMG-HD4 trial and Total Therapy 3 trial. However, the additional treatments in Total Therapy 3 trial are also contained within Total Therapy 2, which means the classifier cannot be specific to those.

We now discuss in more detail in the manuscript that the Total Therapy 3 trial (TT3) contains additional treatments: patients were treated with the regimen VTD-PACE, which includes doxorubicin, dexamethasone, thalidomide, cyclophosphamide, cisplatin and etoposide. The PAD arm of the HOVON-65/GMMG-HD4 also contained doxorubicin and dexamethasone, followed by bortezomib as maintenance therapy. These regimens indeed are documented to impact survival outcomes and the TT3 patients do have a significantly better survival than the patients in the PAD-arm of the HOVON-65 trial. We correct for this survival difference in our Cox-model, so it will not impact the evaluation of the HR in class ‘benefit’ and ‘no benefit’

Changes to the manuscript:

- We have added Supplementary Table 1, which lists which treatments were included in which datasets.

3. The key to the presentation lies in Fig 2; yet the descriptions are confusing and need clarification. From the K-M curves of Fig 2b, it appears Btz benefit and no benefit are the same. Critical to the conclusions is the statement that 19.8% (n=180) show benefits. It is not clear how this number was derived. Further, is the conclusion that the classifier is “showing” benefit or predicts or fits the benefit classifier used in the analysis?

It appears our definition of the ‘benefit’ class was not entirely clear, prompting us to rewrite these parts in the introduction. We define the ‘benefit’ class as a group of patients that has a better survival on bortezomib than on the comparator treatment. We thus expect a difference in the ‘benefit’ class, between the survival curves of the bortezomib and comparator treatment arm; indeed, the ‘benefit’ curve for the comparator treatment is much worse than the bortezomib ‘benefit’ curve (see Figure 2b). STL does not consider the difference between the ‘benefit’ and ‘non-benefit’ classes, since that difference could equally signify a prognostic effect rather than a predictive effect. For aiding future clinical decisions the comparison between treatments within a class ‘benefit’ that we perform is also more useful than a comparison between ‘benefit’ and ‘no benefit’ within one treatment, as we explain in the Introduction section. If no other treatment is considered in the prediction, it is possible that patients who have a bad prognosis still benefit from the treatment; their prognosis could be even worse on another treatment.

Because we define predictive as the difference between treatment within the benefit class, it is perfectly possible for the ‘benefit’ and ‘no benefit’ groups within the bortezomib arm to have

similar survival curves. Moreover, this means the ‘no benefit’ survival curve for the comparator can also be similar, which is indeed the case in Fig 2b.

The 19.8% (n=180 out of n=910) indicates the number of patients that are labeled as class ‘benefit’ (i.e. within the optimal gamma of a prototype in more than T classifiers), i.e. labeled as class ‘benefit’, relative to the total number of patients. This number n=180 is the result of adding up classes ‘benefit’ from Fold D1, D2 and D3. We have clarified this in the results section. We hope these changes make it more clear how we derived our class ‘benefit’ and what we mean when we talk about “showing benefit” .

Changes to the manuscript:

- We changed the initial description of the results to “Figure 2b shows the treatment arms and classes as identified by the STL classifier, when combining the class ‘benefit’ from the three validation folds. These three validation folds together comprise the whole dataset; the classification of each validation fold is predicted by separately trained classifiers. This enables us to show a validation performance for the whole dataset. The validation HRs for the ‘benefit’ and ‘no benefit’ class are 0.50 (p = 0.0012) and 0.78 (p = 0.03), respectively. These results show that a subgroup, comprising 19.8% of the population (n=180 out of 910), is identified by our method that benefits substantially more from bortezomib treatment than the population as a whole; in the entire population an HR of 0.74 (p = 0.0029) is found. More importantly, the STL approach is able to discover and predict this subgroup using the gene expression data at diagnosis.

The bortezomib arm in the ‘benefit’ class and the bortezomib arm in the ‘no benefit’ class have a similar survival, since we predict benefit as compared to the patient group who did not receive bortezomib. As the Kaplan Meier in Figure 2b shows, the other treatment arm in the ‘no benefit’ class also has a similar survival, so we can conclude this group did not benefit from bortezomib; these patients would have had a similar survival had they not received bortezomib. In contrast, the other arm in the ‘benefit’ class has a much worse survival than the bortezomib arm in class ‘benefit’. This group clearly benefits from bortezomib as they survive significantly worse when they do not receive bortezomib (HR bortezomib vs no bortezomib = 0.50, p = 0.0012).”

- We updated the caption of figure 2b to “Kaplan Meier of the combined classifications into a ‘benefit’ and ‘no benefit’ class of D1, D2 and D3. A HR of 0.50 (p = 0.0012) is found between the treatment arms in the ‘benefit’ class and a HR of 0.78 (p = 0.03) in the ‘no benefit’ class. These results show that a subgroup, comprising 19.8% of the population (n=180 out of 910 total), is identified by our method that benefits substantially more from bortezomib treatment than the population as a whole; in the entire population an HR of 0.74 (p = 0.0029) is found.”

4. The authors state “...there are no ground truth labels available...” What is a ground truth label?

A ground truth label refers to a patient characterization known to be accurate. The goal of a typical classification model is to predict a particular patient label as accurately as possible. For instance, ground truth labels are available for datasets used to train prognostic classifiers; based on patient follow-up we can know for sure if a patient survived longer than 5 years or not. For predictive classifiers, which are of interest in our work, such a ground truth label is not available, since we cannot know how a patient would have responded to a different treatment. Since this term may be unfamiliar to more readers, we have elaborated on this.

Changes to the manuscript:

- Changed the sentence mentioning ground truth labels into: “Traditionally, the performance of a classifier is assessed by computing its accuracy, which is done by comparing the predicted labels by the classifier with ground truth labels. Ground truth labels are labels that are known to be accurate because they can be directly observed, e.g. if a patient survives longer than 5 years or not. Since we do not know beforehand which patients benefited from bortezomib, we have no ground truth labels available and cannot compute the accuracy of our classifier.”

5. The authors state that the classifiers capture the same gene patterns, but “they do not agree on class assignment for all patients...” Isn’t this a problem?

In an ideal case, all the classifiers would indeed agree on all the class assignments. The fact that they do not, shows that they are not 100% accurate in predicting treatment benefit: the individual classifiers are making mistakes and these mistakes are random. No published prognostic classifier is 100% accurate and we acknowledge ours is not either. However, related to point 4 of this reviewer, we cannot report an accuracy because the ground truth labels are unknown in this setting.

It is important to note that our classifiers agree on more class assignments than expected by random chance and the classification scores from the different classifiers strongly correlate. This means that we capture a similar signal across the classifiers, indicating that the classifiers agree for the patients that are most confidently classified.

Changes to manuscript:

- Added a clarification to the sentence discussing the overlap between the classifications: “Together, this demonstrates that, even though the classifiers do not agree on the class assignment for all patients (which is expected in practice for classifiers with less than 100% accuracy), they capture the same gene expression patterns.”

6. The authors nicely include analysis of random data sets; but note they can reach significance, just not as good as the STL approach. Still, this seems a problem in overfitting the data used in the classifier modeling.

The random analysis included in the manuscript refers to random gene sets. The survival data and gene expression data are unchanged, and thus the information captured in the gene expression is still intact. What is not available to the classifier, is the biological information contained in the GO (Gene Ontology) sets, since we pooled the genes in random sets. It is not surprising that the classifier can still find a signal, but the superior performance when the GO sets are used demonstrates that the biological information in the GO does help.

Having said this, we think the suggestion to shuffle the treatment labels is a very good one. When doing so, and thus destroying the relationship between gene expression and treatment specific survival in the dataset, no significant HR is observed in the validation, clearly indicating our classifier does not overfit. We have added this result as Supplementary Figure 3.

Changes to the manuscript:

- We have added the paragraph “Control with shuffled treatment labels” to the results section
- We have added Supplementary Figure 3 presenting the results on shuffled treatment labels

The concepts presented are interesting, and may provide some important new insights into better personalized directions for patient treatment. For this reader, the applications were missed because of a need for more clarity in the approaches described in the main text.

We hope the adjustments made to the manuscript based on the reviewer’s comments improved the clarity of the main text.

Reviewer #2 (Remarks to the Author):

The manuscript by Ubels and coworkers uses machine learning to predict the response of individual patients to alternative therapies, based on the patient's gene expression profile. The aim is to use these predictions for personalizing therapy decisions. While numerous approaches exist that successfully predict patient survival based on expression profiles, the question asked here is different: Decide if a given patient will benefit from a specific therapy, or not.

From a machine learning point of view, this is a difficult problem, as in any clinical trial it is only known how a patient has responded to the therapy he or she has actually received, but it is unknown how the patient would have responded to alternative treatment. The authors solve this problem by pooling patients with similar expression profiles, and within these pools compare survival of the subgroups of patients receiving therapy A vs. B. The underlying assumption is that similar gene expression profiles imply similar therapy response.

From a biometrical point of view, the identification of subgroups that benefit from a given therapy (versus those subgroups that do not) is a standard procedure. If a hypothesis exists which marker or combination of markers might define such subgroups, testing if there is a benefit of one treatment over the other in one of the subgroups, but not the other, is a trivial task. The main contribution of the present manuscript is to develop a systematic procedure to test different multivariate, gene-expression based "subgroup" definitions, by taking gene ontology (GO) information into account and defining patient similarity as similarity of expression profiles of gene subsets defined by GO groups.

The authors have done a good job describing their algorithm, and follow a rigorous training procedure based on crossvalidation. From a statistical point of view, there are some methodological flaws which I detail below. The approach presented is novel, and could help personalize cancer treatment. The main flaws I see concern an improper handling of censoring in the analysis that will introduce a bias, the somewhat ad-hoc procedure that requires manual setting (or grid search) for a number of parameters and the lack of an independent replication study to confirm results. I detail these and some additional points below:

- The procedure used to train the "regular" prognostic classifier (fig 1) is flawed and very ad-hoc, and it is no surprise that this is not working. The authors basically compare the top 25% longest surviving patients in the bortezomib group, and compare them against the worst 25% in the "other" treatment group. This mixes two different effects: Survival differences due to other factors, and survival differences due to treatment. It is impossible to attribute any survival differences to one of the two factors with this setup, and in fact they might even cancel each other. Furthermore, the 25% cutoff is very ad hoc, and may not be optimal. Worse yet, the comparison with their predictor is unfair, as only a fraction (50%) of the training data is used for training the "regular" classifier, versus all training data for training of their classifier. Patients that do not do well on Bortezomib and patients that do well with alternative treatment will also contribute important information, and should not be ignored. A final problem comes from the discretization of survival groups, where patients are assigned to "good" or "bad" survival. How are censored survival times handled when identifying the top 25% survivors? If patients with early censoring are ignored, this will introduce a bias in the analysis, as censoring will more affect the "good prognosis" group than the bad prognosis patients. In the extreme case, consider a curative treatment, where all patients receiving the therapy would be censored for progression free survival. The "best" patients may thus be filtered out in the good prognosis group. And this would decrease the difference between the "good" and the "bad" group...

The reviewer proposes a few good suggestions to improve our comparison with a 'regular' prognostic approach. We do not agree with the reviewer that our analysis is flawed, but understand that our description could use clarification to prevent misunderstanding. First of all, we would like to clarify that we do not use only 50% of the data, which indeed would be an unfair comparison. We assign the top and bottom 25% together to class 'benefit', all others are assigned the 'no benefit' label. We also tried different cutoffs than 25% percent (ranging from

5 to 50%), which yielded the same result: those labels could not be predicted by classical approaches. We have added these results as Supplementary table 2, 3 and 4..

The reviewer is right to point out that censoring is not taken into account. We don't expect this to have a large effect, since most patients are censored late: only 22.9% of all censored patients are censored before 3 years and only 9.8% before 2 years. In relation to the point raised by the reviewer that we would be filtering out the best patients by not taking censoring into account, we do not believe this to be the case. Logically, if we would be filtering out a part of the best patients (i.e. not assigning them to class 'benefit'), this would lead to an absence of a significant HR in class 'benefit'. However, when we define class 'benefit' labels as described, we find an HR of 0.02 ($p = < 2 \cdot 10^{-16}$) (or even lower depending on the definition of the class 'benefit'). This approach is a logical and direct way to derive training labels from the treatment and survival data, and similar to how training labels for prognostic labels are often derived, though with the added complication of taking the two treatment arms into account.

However, the above approach did lead to extreme HRs (< 0.1), indicating that these labels are most probably wrong and suggesting the prognostic approach is not useful in this setting. Therefore, we also trained a classifier based on random labelings that yielded an significant HR below 0.5 as calculated by the Cox proportional hazards model. The Cox model does take censoring into account and these labels lead to an HR that is more comparable to the one found in the STL model (Breslow, N. E. (1975). Analysis of Survival Data under the Proportional Hazards Model. International Statistical Review / Revue Internationale de Statistique, 43(1), 45. <http://doi.org/10.2307/1402659>). The Cox model does take censoring into account. The results of this are shown in Table 2 of the Results section: the classifiers based on these labels also did not perform well.

Changes to the manuscript:

- We have extensively revised the section "*Regular classification does not provide accurate prediction of treatment benefit*" to clarify our approach.
- We have added the performance achieved when using other percentages to define the class 'benefit' to the Supplementary Materials.

- The same bias due to censored patients arises in the identification of prototype patients in the first step of the presented "new" algorithm. Can the authors comment on this? How strongly will censoring bias their results?

We would like to stress that censoring is explicitly taken care of in our algorithm. To deal with censoring in the calculation of zPFS, we exclude neighbour pairs for which we cannot be sure who experienced an event first (ie, when they are both censored). On average this leaves 7 neighbours (out of 10 considered) that are used in the zPFS calculation.

Our approach for dealing with censoring by leaving out neighbours for which no conclusion can be drawn is equivalent to the C-index, which is well accepted in the field of survival analyses (Harrell et al. Statistics in Medicine, 1996, Vol 15, 361-387).

Changes to manuscript:

- We have added to the description of Step 1 of our algorithm: “In an approach similar to Harrell’s C-statistic, Δ PFS is only calculated for neighbor pairs where it is clear which patient experienced an event first; if both are censored or one patient is censored before the neighbor experienced an event, Δ PFS is not computed. When $n = 10$ is used, this on average leads to 7 neighbours being used in the calculation of Δ PFS.”

- There are a number of parameters that need to be set in the authors' approach, and it is unclear how they should be optimally chosen and what influence the choice has on results. For example, in step one, the prototype group size is chosen as $n=10$. This parameter will likely have a significant impact, and balances bias against variance of the prototype computations. The effect of the choice of n should be systematically assessed. Parameters k and γ in step 2 and T in step 4 are chosen using grid search, can the authors comment how these parameters influence results?

In response to this point we have performed several analyses to investigate the robustness to settings of the free parameters n , γ , k , and T in more detail.

The parameters k and γ directly determine the classification boundary, and are therefore undoubtedly important. For this reason, they are optimized using an exhaustive grid search which chooses the optimal combination. To investigate how sensitive their precise setting is, we investigated how small changes to the optimal parameters affect the HR found in validation. In essence, a smaller γ leads to a smaller class benefit. We show the effect of changing the γ parameter in two scenarios: leaving all other parameters as is (Supplementary Figure 7) and when also retraining the threshold T which determines how many classifiers need to agree on the ‘benefit’ classification (Supplementary figure 8). We find that the classifier is robust to (small) changes in these parameters, which is a desirable feature for a robust classifier. As can be seen in Supplementary figure 7, when γ decreases, the HR also decreases since a smaller class benefit is identified. This is consistent with our observation that a smaller class benefit leads to a lower HR (Figure 2c). When threshold T is also reoptimized, the HR stays relatively constant when γ is changed, since the threshold T is chosen such that at least 20% of the patients are classified as class ‘benefit’. Supplementary figure 9 shows the number of patients who receive a different class assignment when γ is changed, again with and without reoptimizing threshold T (black line) and with reoptimization (red line, respectively). When threshold T is reoptimized, few patients change classification, showing different settings for γ would identify the same patients as benefitting from bortezomib. We also investigated how sensitive the classifier is to changing the number of genesets in the classifier (with reoptimization of threshold T , Supplementary Figure 10). The red line indicates the validation HR we originally found. As can be seen, there are many settings which achieve a similar or better validation performance, indicating the classifier is also not very sensitive to the exact number of gene sets included. We have also included the training HR found for all combination of γ and k for three of our top performing genesets (Supplementary figure 11 - 13).

Changes to manuscript:

- We have added a Supplementary note, containing the described analysis of robustness. This includes Supplementary figure 7 - 13.
- We added a sentence about the robustness to the results section: “The classifiers are defined by the parameters k and γ and a threshold T . We optimize the combination of k and γ by an exhaustive grid search. However, we find the performance of our classifier is robust to small changes in these parameters (Supplementary note 1).”

- While the authors show good results in a crossvalidation setting on two multiple myeloma datasets (one focusing on Bortezomib, the other on Lenalidomide treatment), the validation of the predictive models is done on the same datasets (using a hold-out validation set). There are numerous studies that show that performance on completely independent data sets will decline, hence a further, completely independent dataset would be very useful for model assessment. The ideal situation would of course be a prospective study assessing the performance of the model in assigning treatments over a treatment decision according to guidelines. Clearly, this is not feasible; however, could other MM/Bortezomib studies be used to at least perform a validation on an independent data set? If this also is not feasible, what would happen if the authors train their predictor on the TT3 data set and make predictions on the H65 data?

We are aware of the fact that cross-study validation often shows that some fitting to the dataset at hand has occurred. This is why it is important to ensure that the classifier is trained on data with sufficient heterogeneity, which is what we aimed for when pooling the TT and H65 data. Unfortunately, additional datasets that can be used to further increase training set heterogeneity or serve as additional independent validation set are lacking because, after the Total Therapy 3 and H65 trials were successfully completed, bortezomib became the most common drug to treat MM and thus there is no data publicly available for patients who did not receive bortezomib. A trial containing similar treatment arms was conducted in Italy (GIMEMA MMY-3006), but unfortunately only the bortezomib arm was profiled by microarray.

Prompted by the suggestion from this reviewer, we have further investigated the validation results of our classifier. To this end, we split the validation folds into the two datasets (i.e. H65 and TT) and calculated two validation HRs, one for each dataset. Reassuringly, we find that for both datasets the expected effect is observed - albeit not significant due to small sample size (Supplementary figure 5 and 6).

We also performed a new round of cross-validation this time only using the TT dataset. This classifier performs substantially better (i.e. achieved a lower HR) than the classifier trained on mixed data.

We also followed the suggestion by the reviewer to train on the Total Therapy datasets and validate on H65 (the other way around is not possible due to small sample size of the H65 set). This classifier did not perform well in validation, corroborating our observation that dataset specific fitting has occurred. We hypothesize this is because the two datasets are very different both in population and treatment regimes. The Total Therapy trial included far more extensive treatment combinations than H65 trial (see Supplementary table 1 for details). When the STL classifier is trained on the TT dataset only, it is very well possible we are deriving a classifier specifically for the TT treatment combination, rather than bortezomib. Moreover, the TT trials were conducted in the USA, where the H65 trial was conducted in Europe, leading to different study populations.

We corrected for batch effects in the data by applying ComBat. After this correction we could not find significant differences between the dataset using a t-test or separate them with a nearest mean classifier. However, further inspection revealed that both datasets can be separated with 85% accuracy when using a random forest classifier. Rather than batch effect, this could also be a true biological difference between the American and European study populations. As a result, it may not be surprising that when we train on only Total Therapy the classifier dataset specific effects are important to reaching a good classifier.

These results indicate that it is very important to match the training population with the population one intends to use the classifier in.

We have added a discussion on data heterogeneity and the above observations in the results section.

Changes to the manuscript:

- We have added a section “Impact of dataset of origin on validation performance” to the results section that further addresses this issue and shows the performance when we train on the more homogeneous Total Therapy dataset.

- Along those lines, as the authors claim that their approach is generalizable, I feel a second disease should be chosen and similar analyses be shown.

While we do not expect the approach to be specific to Multiple Myeloma, the reviewer is correct that we have not shown STL is generalizable to other diseases. We would like to stress that we have applied the STL approach to another MM dataset that was generated on a different platform and which stratified according to a different drug, showing that our approach is at least not dataset specific. We have added a comment about this to the discussion.

Changes to the manuscript:

- Changed the relevant paragraph in the discussion section to: “The STL classifier was successfully tested across different gene expression platforms, different treatments and different study types, demonstrating that STL is more generically applicable than one particular dataset. Since our work has focused on MM, an important next step is to

investigate if STL is also successful in unraveling treatment benefit for other diseases. If so, STL can play an important role in rescuing treatments that do not achieve a significant effect in the entire patient population but may still benefit a subset of the patients.”

- GO contains hierarchical gene groups. Is this taken into account in the algorithm presented? If not, the analysis of how often a gene occurs in different predictors may simply be due to a larger number of hierarchy levels for that gene, and may not give any indication of "biological importance"

The reviewer is right that not every gene is annotated to the same number of categories or hierarchy levels in GO. However, we have taken this into account; the significant overrepresentation that is mentioned in the results section is based on how often the gene is chosen versus how often we would expect to see it based on all the GO terms to which it is annotated. We described this in the methods section under **Calculating overrepresentation of genes used in the classifier**.

Changes to the manuscript:

- We added an additional explanation to the results section “*Genes used to predict bortezomib benefit*”: “GO sets are hierarchical (i.e. there is a larger parent category that can include several children categories) and genes can be annotated to multiple GO categories. Therefore we have taken into account how many GO categories include a certain gene to establish if we observe a gene more often than expected in our classifiers.”

- Please provide confidence intervals for computed Hazard Ratios and list the exact test used along with n for all reported p-values; clarify for all computed hazard ratio whether the ratio is computed as A/B or B/A. Furthermore, please provide p-values for all computed HRs, e.g. in Table 1

We have added the requested details to the tables and text.

Changes to the manuscript:

- We added to the “*Endpoint and survival analysis*” section: “Hazard Ratios (HR) and associated 2-sided p-values were calculated. P-values below 0.05 were considered statistically significant. All HRs are computed as bortezomib vs no bortezomib and lenalidomide vs no lenalidomide, which means an HR below 1 signifies a benefit when receiving bortezomib or lenalidomide.”
- We added p-values to Table 1 and Table 2, as well as the newly added Supplementary tables 2, 3 and 4.
- We added the values of n for all the listed HRs

- Please provide more details on the Multiple Myeloma data sets used, in particular characteristics of patients (age, gender, immunoglobulin subtype, status of genetic markers such as 17p, etc)

We have added a description of the age and gender distribution to the methods section, which is all the metadata that is available to us. We do not have FISH data available for the entire dataset. The analysis included in the results section, which compares the STL performance to FISH markers and molecular subtype, was based on so-called virtual FISH, which predicts cytogenetics from gene expression data.

Changes to the manuscript:

- We have added additional details to the description of the datasets in the Methods section: “The TT2 dataset included 345 newly diagnosed multiple myeloma (NDMM) samples, treated either with thalidomide and melphalan (n = 173) or melphalan alone (n = 172). Average age is 56.3 (range: 24 - 76) and 57.1% of the patients is male. The TT3 dataset included 238 NDMM samples treated with bortezomib, thalidomide, dexamethasone, cyclophosphamide, cisplatin and etoposide (VTDPACE). Average age is 58.7 (range: 32 - 75) and 67.6% is male. The H65 dataset included 327 NDMM samples, treated either with vincristine, doxorubicin and dexamethasone (VAD, n = 158) or bortezomib, doxorubicin and dexamethasone (PAD, n = 169). Average age is 54.7 (range: 27 - 65) and 56.4% percent is male.”

- The authors explain the difference between what they call "prognostic" and "predictive" signatures in Figure 1a and b. They define "prognostic" signatures as signatures that do not take treatment into account in the text, but the example they show in Figure 1a as a "prognostic" model is a classifier that is trained to distinguish the patients with best survival in the treatment group against the 25% worst survivors in the control group. This is inconsistent with their previous definition.

It seems we were not clear enough about how we train the regular classifier, which we believe may have prompted this question by the reviewer. The classification shown in Figure 1a is indeed prognostic and no significant HR is observed between the treatment arms in the good survival group. The classifiers we train in the section “*Regular classification does not provide accurate prediction of treatment benefit*” are trained to be predictive (and not prognostic as in Figure 1a), but they do not validate. Hence the need for the algorithm we present in this paper. We hope and expect that our changes in response to this reviewers’ comment about comparison with regular classifiers also clears this up. We assign the top (bortezomib) and bottom (non-bortezomib) 25% together form class ‘benefit’, all others (75% of the patients) are assigned the ‘no benefit’ label. This is not inconsistent with our definition of predictive.

Changes to the manuscript:

- We have extensively revised the section “*Regular classification does not provide accurate prediction of treatment benefit*” to clarify our approach.

- I did not understand supplementary figure 1. Could you please explain in more detail what is shown and what the conclusions are?

We have changed the description of Supplementary Figure 1 to clarify.

Changes to the manuscript:

- We have changed the caption of Supplementary Figure 1 to : “We computed for how many patients the three classifiers trained in the different folds of the cross validation agree on class assignment. The values on the x-axis represent the number of classifiers that classified a patient as benefitting from treatment. A value of 0 means that all three classifiers classified a patient as ‘no benefit’ and the value of 3 (which is the maximum) means all classifiers agreed on the assignment to class ‘benefit’. These are the red dots in the plot. We also generated 10 000 random labelings per training fold, with the same proportion of patients labeled ‘benefit’ and ‘no benefit’ as in the labelings found by STL to obtain a background distribution of the expected overlap by random chance (boxplot). Since the number of patients for which all three STL classifier agree (i.e. the patients with either a value of 0 or 3) is larger than expected by random chance, the concordance between the STL classifiers is significant.”

REVIEWERS' COMMENTS:

Reviewer #1 (Remarks to the Author):

The authors have done a very detailed and thorough response to all comments.

I do wonder if this analysis could be applied to other proteasome inhibitors as validation for which there would be additional data sets. It would be of significant interest if this approach could provide benefit analysis across proteasome inhibitors.

Reviewer #2 (Remarks to the Author):

I thank the authors for their detailed response and the additional computations they have performed. All my comments have been addressed appropriately.

As Reviewer 2 did not offer additional comments on our manuscript, we respond to reviewer 1 only. We have copied the additional comment from reviewer 1 below in blue, with our response in black.

Reviewer #1 (Remarks to the Author):

The authors have done a very detailed and thorough response to all comments.

I do wonder if this analysis could be applied to other proteasome inhibitors as validation for which there would be additional data sets. It would be of significant interest if this approach could provide benefit analysis across proteasome inhibitors.

We agree it would be very interesting to evaluate the performance of our classifier in another proteasome inhibitor dataset, to investigate if we capture a bortezomib-specific or a proteasome inhibitor-specific effect. Unfortunately, the only other proteasome inhibitor that is (widely) used in Multiple Myeloma is carfilzomib. Carfilzomib was introduced into the clinic quite recently and no mature datasets exist, which prevents us from investigating this question at this time.